# The Effect of Disulfiram and Copper on Cellular Viability, ER Stress and ALDH Expression of Human Meningioma Cells

**DOI:** 10.3390/biomedicines10040887

**Published:** 2022-04-12

**Authors:** Ying Kao, Li-Chun Huang, Shao-Yuan Hsu, Shih-Ming Huang, Dueng-Yuan Hueng

**Affiliations:** 1Graduate Institute of Medical Sciences, National Defense Medical Center, Taipei 11490, Taiwan; dxr60@tpech.gov.tw; 2Division of Neurosurgery, Department of Surgery, Taipei City Hospital Zhongxing Branch, Taipei 10341, Taiwan; 3Department of Biochemistry, National Defense Medical Center, Taipei 11490, Taiwan; emily7781@hotmail.com (L.-C.H.); shihming@ndmctsgh.edu.tw (S.-M.H.); 4Division of Neurosurgery, Department of Surgery, Taipei City Hospital Renai Branch, Taipei 106243, Taiwan; daw94@tpech.gov.tw; 5Department of Neurological Surgery, Tri-Service General Hospital, National Defense Medical Center, Taipei 11490, Taiwan

**Keywords:** DSF, disulfiram, copper, meningioma, meningioma stem-like cells, ER stress, ALDH

## Abstract

(1) Background: Meningiomas are the most common intracranial tumors in adults; currently there is no effective chemotherapy for malignant meningiomas. The effect of disulfiram (DSF)/Copper (Cu) on meningiomas remains unclear; (2) Methods: The impact of DSF/Cu on cell viability of meningioma adhesion cells (MgACs) and sphere cells (MgSCs) was assessed via MTS assay. The effects of DSF/Cu on intracellular Cu levels, cell senescence, and apoptosis were analyzed using CopperGreen, C_12_FDG, and Annexin V assays. Intracellular ALDH isoform expression and canonical pathway expression after DSF/Cu treatment were analyzed using mRNA microarray and Ingenuity Pathway Analysis, with further verification through qRT-PCR and immunoblotting; (3) Results: The viability of MgACs and MgSCs were inhibited by DSF/Cu. DSF/Cu increased intracellular Cu levels and cellular senescence. DSF/Cu also induced ER stress in MgACs and activated the PERK/eIF2 pathway for further adaptive response, apoptosis, and autophagy. Finally, DSF/Cu inhibited the expression of different ALDH isoforms in MgACs and MgSCs; (4) Conclusions: DSF/Cu exerts cytotoxic effects against both meningioma cells and stem-like cells and has treatment potential for meningioma.

## 1. Introduction

Meningiomas mainly arise from arachnoid cap cells that are located at the surface of the brain parenchyma, though they are occasionally found at intraosseous or intraventricular locations. They are the most commonly occurring intracranial tumor, with an incidence rate of 7.89 per 100,000 people per year. According to data from the Central Brain Tumor Registry of the United States (CBTRUS), meningiomas account for 35% of primary brain tumors and occur about twice as often in females as males [1]. Nearly 80% of meningiomas are benign and classified as WHO grade I. In addition, about 80% of meningiomas are surgically curable. Subsequent recurrence of the tumor will depend on the degree of tumor resection [2,3] and whether the tumor location allows for Simpson grade I resection. The behavior of WHO grade II or III meningiomas (e.g., atypical meningiomas or anaplastic meningiomas) is more aggressive and invasive, which correlates with a higher recurrence rate. According to Moliterno et al., overall survival among patients with anaplastic meningioma is 2.7 years, with 2- and 5-year survival rates of 66.6% and 27.9%, respectively. In that study group, about 70% of patients received more than two craniotomies during the treatment course [4]. According to the National Comprehensive Cancer Network (NCCN) guidelines, re-operation, radiation therapy, and chemotherapy are suggested in the event of tumor recurrence [5]. However, in patients with surgery- and radiation-refractory meningiomas treated with chemotherapy, the 6-year progression-free survival rate among patients with WHO grade I tumors is only 29% and is only 26% among those with WHO grade II/III tumors [6]. Thus, new and effective therapies are urgently needed to improve the outcome of patients with aggressive meningiomas.

Disulfiram (C_10_H_20_N_2_S_4_; tetraethyl thiuram disulfide, Antabuse, DSF) [7] has been used to treat alcoholism since 1951, as it also chelates copper (Cu) ions with its thiol group [7]. In recent studies, DSF complexed with Cu (DSF/Cu) was effective against various cancers. Iljin et al., found that DSF/Cu inhibited the growth of prostate cancer cells [8], while Chen et al., reported that DSF/Cu induced apoptosis in breast cancer cells through proteasome inhibition [9]. DSF/Cu induces cell apoptosis in lymphoma by activating the JNK pathway and inhibiting NF-κB and Nrf-2 [10]. Within the body, DSF can be metabolized to ditiocarb, and the ditiocarb/Cu complex inhibits the p97 segregase adaptor NLP4, which alters the metabolic pathway of breast cancer cells, ultimately inhibiting cancer growth [11]. Moreover, DSF/Cu is also reported to effectively inhibit the growth of cancer stem cells [12].

DSF reportedly inhibits the growth of glioma stem cells [13] and temozolomide-resistant glioblastoma cells [14]. DSF/Cu inhibits O6-methylguanine-DNA methyltransferase in human glioblastoma (GBM) cells, increasing DNA-alkylating damage [15]. Lun et al., reported that DSF/Cu increases glioblastoma cells’ radio- and chemosensitivity by inhibiting the DNA repair [16]. Huang et al., reported that glioblastoma patients treated with temozolomide in combination with DSF had progression-free survival of 8.1 months [17], which is better than the outcomes of the Stupp protocol, RTOG 0525, and RTOG 0825 [18,19,20].

The aldehyde dehydrogenases (ALDHs) family contains 19 human isozymes. ALDH is critical in forming molecules such as retinoic acid, γ-aminobutyric acid, and betaine, which are essential in maintaining cellular homeostasis [21,22]. Also, ALDH plays a vital role in the self-renewal of stem cells and cancer stem cells by conversing the toxic aldehydes into non-toxic carboxylic acids. As a marker of cancer stem cells, ALDH activity is related to the clonogenicity and tumorigenicity of cancer cells [23]. ALDH expression is related to drug resistance and radioresistance of tumor cells and is associated with poor outcomes in different kinds of cancers [24,25,26,27,28,29,30,31].

Tumor stem-like cells are critical in tumor recurrence, angiogenesis, and invasion. It would be meaningful to use meningioma cells and stem-like cells to discover the potential treatment strategy of malignant meningiomas. In the present study, we used meningioma cells (Meningioma Adhesion Cells, MgACs) and meningioma stem-like cells (Meningioma Sphere Cells, MgSCs) [32] (Figure 1) to investigate the antitumor effects of DSF/Cu. Here we characterize the impact of DSF/Cu and identify its different target mechanisms in these two cell types.

## 2. Materials and Methods

### 2.1. Cell Culture, RNA Isolation, and Quantitative RT-PCR

Prof. Hueng established the P9 MgACs and MgSCs from brain tumor specimens of one individual patient (P9 means Patient #9). For meningioma cells isolation, the specimen was washed in phosphate-buffered saline (PBS) twice to remove blood, chopped into small pieces, and digested in collagenase type IV (2 mg/mL in DMEM) for 30 min at 37 °C. The cells were washed with PBS twice and cultured in Dulbecco’s modified Eagle medium (DMEM) (Corning, Glendale, AZ, USA) supplemented with 10% fetal bovine serum (FBS). Since the nature of tumor cells is heterogenicity and there is a limitation of daily case numbers for craniotomy surgery, we did not routinely undergo pooling of samples from several donors. The primary cell characterization was performed. In brief, the tumor cells revealed positive human epithelial membrane antigen (EMA) and vimentin in histopathological staining. To obtain MgSCs sphere cultures, we maintained MgACs in stem cell media consisting of DMEM/F12 (Corning, Glendale, AZ, USA) supplemented with 1X N2 (Thermo Fisher Scientific, Waltham, MA, USA), 20 ng/mL epidermal growth factors (EGF, PeproTech Asia, Israel), and 20 ng/mL fibroblast growth factors (FGF, PeproTech Asia, Israel), to form the sphere cells (MgSCs) as previously described. The characterization and function of MgSCs, such as self-renewal, differentiation potential, and stem-like cell markers, were confirmed [32]. Meningioma cells from primary culture were further collected from three other different patients (meningioma #1, #2, and #3). The IOMM-Lee cell line was purchased from American Type Culture Collection (ATCC). The meningioma #1, #2, and #3; MgACs; and IOMM-Lee cells were cultured in Dulbecco’s modified Eagle medium (DMEM) (Corning, Glendale, AZ, USA) supplemented with 10% fetal bovine serum (FBS). Disulfiram (DSF), CuCl_2_, CuSO_4_, FeCl_3_, CoCl_2_, MgSO_4_, NiSO_4_, and MnCl_2_ were purchased from Sigma-Aldrich (Burlington, MA, USA); CuCl was from Alfa Aesar (Tewksbury, MA, USA). CuCl was dissolved in H_2_O, while all other drugs were dissolved in DMSO to generate stock solutions. Total RNA was extracted using EasyPure Total RNA reagent according to the manufacturer’s protocol (Bioman, Taipei, Taiwan). For cDNA synthesis, 1.0 μg of RNA was reverse transcribed into cDNA using Oligo dT primer with MMLV Reverse Transcriptase (Epicentre Biotechnologies, Madison, WI, USA). Normal brain cDNA was obtained from Origene Technologies (Rockville, MD, USA). Gene expression was quantified using quantitative RT-PCR (qRT-PCR) with a StepOne™ Real-Time PCR System (Thermo Fisher Scientific, Waltham, MA, USA). Amplifications were performed using an IQ2 fast qPCR system with ROX (Bio-genesis Technology Inc., Taipei, Taiwan). Quantification of gene expression relative to internal control, GAPDH, was done using the 2^−ΔΔCt^ method [33]. The primer pairs used are listed in Table 1.

### 2.2. Cell Lysate Preparation and Western Blotting

Cells were lysed in RIPA buffer (100 mM Tris-HCl, 150 mM NaCl, 0.1% SDS, and 1% Triton-X-100) for 10 min at 4 °C, after which the lysed cells were centrifuged at 15,000 rpm for 10 min, and the supernatants were collected as the cell lysates. Thirty-microgram aliquots of cell lysate from each group were applied to 10% sodium dodecyl sulfate-polyacrylamide gel electrophoresis. Proteins were then transferred onto polyvinylidene fluoride membranes (Millipore, MA, USA) and blocked with 5% skim milk in TBST for 1 h at room temperature. Antibodies were purchased from Cell signaling Technology (Danvers, MA, USA) (cyclin B1 and p21), Santa Cruz Biotechnology (Dallas, TX, USA) (ATF-3, ATF-4, GRP78, ALDH1A2, ALDH1L1, ALDH4A1, and GADPH), Novus Biologicals (Centennial, CO, USA) (ATF-6 and XBP1), Cell Signaling Technology (Danvers, MA, USA) (CHOP, cleaved PARP, Bax, cleaved caspase 3, cleaved caspase 7, caspase 9, and LC3), Abcam (Cambridge, MA, USA) (γH2A.X, ALDH 1A2, ALDH 1A3, ALDH 3B1, ALDH 16A1, and cyclin D1), and MB and Atlas Antibodies AB (Bromma, Sweden) (ALDH 1B1). Antibodies were diluted at the indicated ratio with enhancer kits following the protocol of the manufacturer (ATF-3: 1:500 in Ab Enhancer Buffer; ATF-4: 1:1000 in Ab Enhancer Buffer; GRP78: 1:2000 in milk; ALDH1A2: 1:1000 in Ab Enhancer Buffer; ALDH1L1: 1:3000 in milk; ALDH4A1: 1:1000 in milk; ATF-6: 1:1000 in Ab Enhancer Buffer; XBP1: 1:1000 in Ab Enhancer Buffer; CHOP: 1:800 in Ab Enhancer Buffer; cleaved PARP: 1:1000 in Ab Enhancer Buffer; Bax: 1:1000 in Ab Enhancer, cleaved caspase 3: 1:500 in Ab Enhancer; cleaved caspase 7: 1:1000 in Ab Enhancer; caspase 9: 1:1000 in Ab Enhancer; γH2A.X: 1:1000 in 1% BSA; GADPH: 1:10000 in 1% BSA; LC3: 1:2000 in Ab Enhancer; ALDH 1A2: 1:1000 in milk; ALDH 1A3: 1:1000 in milk; ALDH 3B1: 1:1000 in milk; ALDH 16A1: 1:1000 in milk; ALDH 1B1: 1:2000 in milk; cyclin D1: 1:1000 in Enhancing Buffer; p21: 1:1500 in milk; cyclin B1: 1:1000 in Enhancing Buffer). Bands were detected using enhanced chemiluminescence and X-ray film (GE Healthcare, Piscataway, NJ, USA).

### 2.3. Cell Viability

To assess cell viability, meningioma #1 and #2 were plated at a density of 5 × 10^3^ cells/well, meningioma #3 at 3 × 10^3^ cells/well, MgACs at 3 × 10^3^ cells/well, MgSCs (using MgACs cultured in stem cell media) at 2 × 10^4^ cells/well, and IOMM-Lee cells at 5 × 10^3^ cells/well in 96-well plates. The next day, the cells were treated with DSF alone and combined with CuCl, CuCl_2_, CuSO_4_, FeCl_3_, CoCl_2_, MgSO_4_, NiSO_4_, or MnCl_2_ for the indicated times. Twenty microliters of MTS reagent was then added for 2–4 h according to the manufacturer’s protocol (CellTiter 96 Aqueous One Solution Cell Proliferation Assay, G3581, Promega, Madison, WI, USA), after which the absorbance in 490 nm was measured using Varioskan LUX multimode microplate reader (Thermo Scientific, Thermo Fisher Scientific, Waltham, MA, USA) [34].

### 2.4. Cell Apoptosis, Senescence, Intracellular Cu Analysis, and Flow Cytometric Analysis

Annexin-V assays were used to analyze cell apoptosis. Cells were seeded into 6-well plates and treated with DSF/Cu the next day. The cells were then collected and processed using the protocol in the manufacturer’s guide (#559763, BD Biosciences, Franklin Lakes, NJ, USA). Apoptosis was assessed using a BD FACSCalibur flow cytometer (BD Biosciences, Franklin Lakes, NJ, USA) [35,36,37,38].

For flow cytometric analysis of cell senescence, the cells were stained with 5-Dodecanoylaminofluorescein Di-β-D-Galactopyranoside (C_12_FDG, D2893, Invitrogen, Thermo Fisher Scientific, Waltham, MA, USA), a substrate that becomes fluorescent and membrane impermeant when cleaved by β-galactosidase. MgACs and MgSCs were trypsinized with trypsin EDTA, washed with PBS, and incubated for 1 h in 50 μL of PBS with 33 μM C_12_FDG. The cells were then immediately analyzed using a FACSCalibur flow cytometer. The data obtained were analyzed using FACSDiva software (Version 6.1, BD Biosciences, San Jose, CA, USA). The fluorescein signal from C_12_FDG was measured on an FL1 detector. The activity of β-galactosidase was estimated as the median fluorescence intensity (MFI) of the meningioma [39,40].

Intracellular Cu levels were estimated using CopperGREEN, a Cu^+^-selective dye (GORYO Chemical #GC902, Sapporo, Japan). After treating cells with DSF/Cu for 24 h or 48 h, cells were rinsed with PBS containing 200 μM of ethylenediaminetetraacetic acid (EDTA, Thermo Fisher Scientific, Waltham, MA, USA) to remove extracellular copper ion. Then, cells were incubated with 5 μM of copperGREEN diluted in a cell culture medium for 3 h. The fluorescence was then analyzed using a BD FACSCalibur flow cytometer (BD Biosciences) [41,42].

### 2.5. UV-Vis Spectral Analysis

We prepared CuCl and CuCl_2_ in an aqueous solution with a concentration of 1 mM and then added them into DeNOVIX DS-11 FX+ Spectrometer to detect the absorption.

### 2.6. RNA Microarray Analysis

Total RNA was extracted from MgACs using an RNeasy Mini kit (Qiagen, Valencia, CA, USA) and analyzed with a 2100 Bioanalyzer (Agilent Technologies, Palo Alto, CA, USA) to evaluate whether the RNA quality met the requirements of the microarray chip. RNA was quantified and then applied to a GeneChip 3′ IVT Expression Kit & Hybridization Wash and Stain Kit (Affymetrix, Santa Clara, CA, USA). The RNA was loaded onto Affymetrix GeneChip Human Genome U133 Plus 2.0 arrays (Affymetrix, Santa Clara, CA, USA) chip. All gene-related probes were analyzed and Log2-standardized with GeneSpring (Agilent Technologies, Palo Alto, CA, USA). We further analyzed gene probes with >1.3-fold and canonical ingenuity pathways with >1.0-fold increase/decrease in MgACs with DSF/Cu treatment compared with the cells with DMSO treatment (control) only in the control group (*n* = 1). We confirmed that cells with DSF treatment versus control and DSF/Cu treatment versus control had consistent changes. Furthermore, we predicted the potential upstream transcription factors and signaling pathways with Ingenuity Pathway Analysis (IPA, QIAGEN, Redwood, CA, USA) online tools [43,44,45]. These microarray data were uploaded to the National Center for Biotechnology Information (NCBI) Gene Expression Omnibus (GEO, GSE196689).

### 2.7. Statistical Analysis

Student’s *t*-test or one-way analysis of variance were used for comparisons between groups. The results are presented as the means ± SD or as specified. Values of * *p* < 0.05, ** *p* < 0.01, and *** *p* < 0.001 were considered significant.

## 3. Results

### 3.1. DSF/Cu Reduces Meningioma Cell Viability

To assess the effect on cell viability of DSF alone and DSF/Cu, we treated meningioma primary culture cells, IOMM-Lee cells, MgACs, and MgSCs for 48 h with various concentrations of DSF with or without CuCl_2_. We then assessed cell viability with MTS assays (Figure 2). We found that the viability of meningioma #1, #2, and #3 (Figure 2a); IOMM-Lee; MgACs; and MgSCs (Figure 2b) were all reduced by DSF/Cu. Moreover, Cu^2+^ was not as cytotoxic alone as in combination with DSF. The DMSO used to make the stock solutions show lesser toxicity than DSF/Cu (Figure 2c–e). These findings suggest that DSF/Cu exerts an inhibitory effect on the viability of both meningioma cells and meningioma stem-like cells.

### 3.2. DSF Combine with Cu^+^ or Cu^2+^ Reduces the Viability of Meningioma Cells

Because DSF is a copper chelator [7], we wanted to know the effect of the valence of the copper ion on the antitumor effect of the DSF/Cu complex. First, we conducted the UV-Vis spectral analysis of CuCl 1 mM and CuCl_2_ 1 mM in an aqueous solution. As shown in Figure 3a, the absorbance spectra of CuCl and CuCl_2_ were 260~280 nm and 190~200 nm, separately [46,47]. Then, we treated MgACs and MgSCs for 48 h with DSF/CuCl, DSF/CuCl_2_, or DSF/CuSO_4_. As shown in Figure 3, The viability of both MgACs (Figure 3a) and MgSCs (Figure 3b) was inhibited by DSF complexed with Cu^+^ or Cu^2+^, indicating that DSF/Cu is cytotoxic to meningioma cells and stem-like cells, irrespective of the valence of the copper ion. In addition, similar results obtained with DSF/CuCl_2_ and DSF/CuSO_4_ indicate that the corresponding anion has little or no effect.

### 3.3. DSF in Complex with Different Metal Ions Showed Little or no Cytotoxic Effect in Meningioma Cells

Mello Filho et al., suggested intracellular iron-induced cell death through the Fenton reaction [48]. Copper, chromium, and cobalt are also reportedly involved in cytotoxic Fenton-like reactions [49]. In addition to copper, DSF also chelates nickel [50]. To assess the specificity of the DSF/Cu cytotoxicity, we evaluated the viability of MgACs and MgSCs after incubation for 48 h with DSF/Fe, DSF/Co, DSF/Mg, DSF/Ni, or DSF/Mn. The results show that DSF in complex with Fe, Co, Mg, Ni, or Mn did not suppress cell viability as DSF/Cu did (Figure 4). Thus, the cytotoxic effect of the DSF/Cu complex appears to be copper-specific.

### 3.4. DSF/Cu Induces Intracellular Copper Accumulation and Cell Senescence in Meningioma

To investigate whether DSF/Cu causes a change in the intracellular Cu^+^ concentration in meningioma cells, we used CopperGREEN as a probe with flow cytometry. We found that intracellular Cu^+^ levels were increased in MgACs and MgSCs after 24 h in the presence of DSF/Cu. After 48 h of treatment, DSF/Cu^1+^ and DSF/Cu^2+^ resulted in more obvious increased intra-cellular Cu concentration in MgSCs than in MgACs (Figure 5a). Further, we wanted to know if DSF/Cu could induce cell senescence, so we performed a C_12_FDG assay. The data showed that DSF/Cu increases the cellular senescence among MgACs and MgSCs especially after 48 h treatment (Figure 5b). Taken together, these results indicate that DSF/Cu induces cellular copper accumulation and cellular senescence in meningioma cells.

### 3.5. DSF/Cu Induced ER Stress in Meningioma Cells

We next extracted the mRNA from MgACs treated with DSF/Cu for 6 h or 24 h and performed microarray analysis. Using Ingenuity Pathway Analysis (IPA), we found that transcription factors ATF3, ATF4, ATF6, and XBP1, which all play roles in the unfolded protein reaction, were activated after DSF/Cu treatment (Figure 6a,b). Immunoblotting revealed that the expressions of ATF3, ATF4, and CHOP, which are all involved in the PKR-like ER kinase (PERK) pathway, were elevated after DSF/Cu treatment, and the change was more evident in MgACs than MgSCs (Figure 6c). For a possible canonical pathway affected by DSF/Cu analyzed via IPA, we found that the eIF2 signaling pathway revealed upregulation after 6 h of treatment and was downregulated after 24 h treatment of DSF/Cu (Figure 6d,e). We further evaluated the proteins related to this pathway. The immunoblotting revealed that after 6 h of treatment, the levels of ATF4, p21, and cyclin D1 were increased in MgACs. After 24 h of treatment of DSF/Cu, the levels of p21 and cyclin D1 were decreased, and the level of CHOP was increased in MgACs compared with the 6 h group (Figure 6f). However, the phenomenon was not evident in MgSCs. Based on these results, we think that DSF/Cu-mediated increases in the intracellular Cu concentration further induced ER stress in meningioma cells. In addition, the PERK pathway, which is a branch of the unfolding protein response, was activated. Furthermore, we think that via the PERK/eIF2 pathway, the cell cycle is arrested in the G1 phase during short-term ER stress for the adaptive response. However, when ER stress is prolonged, the amount of ATF4 and CHOP increases, leading to cell senescence and apoptosis.

### 3.6. The Effect of DSF/Cu on Cell Apoptosis and Autophagy in Meningioma Cells

To further explore the apoptosis associated with ER stress, we conducted the Annexin V assay, and the result showed that DSF/Cu increases cell apoptosis among meningioma cells (Figure 7a). Moreover, the immunoblotting of the proteins related to apoptosis revealed that cleaved caspase 7 and γH2A.X increased in MgACs and MgSCs after DSF/Cu treatment. Furthermore, Bax, cleaved caspase 3, and procaspase/cleaved caspase 9 were not different with or without the DSF/Cu treatment (Figure 7b). We further investigated the effect of DSF/Cu on cellular autophagy. The data revealed that DSF/Cu induced autophagy in meningioma cells (Figure 7c). Our data showed that DSF/Cu induces apoptosis and autophagy in meningioma cells.

### 3.7. DSF/Cu Inhibits Different ALDH Isoforms Expression in Meningioma and Stem-like Cells

To identify the ALDH isoforms affected by DSF/Cu, we used a gene microarray to survey the gene expression of ALDH isoforms in DSF/Cu-treated MgACs. We found that compared to the control (DMSO), the expression of ALDH 1A2, 1A3, 1B1, and 4A1 was decreased after 6 h of DSF/Cu treatment. After 24 h of DSF/Cu treatment, the expression of ALDH 1A2, 1A3, 1L1, 3A1, 3A2, 3B2, 4A1, and 16A1 was decreased (Figure 8a). These results were validated and confirmed by qRT-PCR. The expression of ALDH 1A2, 1A3, 1B1, 3A1, 3A2, and 4A1 was decreased after DSF/Cu treatment in MgACs. In MgSCs, the expression of ADLH 1A2, 1A3, 1B1, and 16A1 was decreased after DSF/Cu treatment (Figure 8b). Moreover, immunoblotting showed that the protein expression of ALDH 1A2 and 1A3 was decreased in MgACs under DSF/Cu treatment, and the protein and the levels of ALDH 1A2, 1A3, 1B1, 1L1, 4A1, and 16A1n were decreased in MgSCs after DSF/Cu treatment (Figure 8c). However, the decreased protein level of ALDH 1A2 and 1A3 revealed not as obviously in MgACs as in MgSCs. This suggests the influence of DSF/Cu on different ALDH isoforms’ expressions in MgACs and MgSCs.

## 4. Discussion

This is the first study to investigate the treatment effects of DSF/Cu on meningioma cells. Our results suggest that DSF/Cu can reduce the survival of meningioma cells and stem-like cells. DSF/Cu can increase the intracellular copper concentration, ER stress, and cell apoptosis. We also determined that DSF/Cu inhibited the expression of different ALDH isoform in MgACs and MgSCs.

The Fenton reaction was first reported by H.J. Fenton in 1894, who described the oxidation of tartaric acid in the presence of iron [51]. In humans, the Fenton reaction results in the production of hydroxyl ions are as follows.
Fe^2+^ + H_2_O_2_ → Fe^3+^ + •OH + OH^−^(1)

Moreover, it was later shown that the hydroxyl ion produced in this reaction damages DNA, resulting in cell death [48]. More recently, it was reported that copper toxicity involves a Fenton-like reaction [52] and that other metal ions, including Co and Cr, can also cause DNA damage via redox reactions [49]. Our results show that DSF complexed with Cu is cytotoxic in MgACs and MgSCs, irrespective of the Cu valence. However, DSF was less cytotoxic when combined with other metal ions, including Fe, Co, Mg, Ni, and Mn. Therefore, we suggest the Fenton reaction may not be the primary mechanism by which DSF/Cu suppresses meningioma cell survival.

Zhang et al., reported that DSF/Cu induces ER stress via an IRE1α-XBP1 pathway leading to autophagy-dependent apoptosis in pancreatic and breast cancer cells [53]. Our data show that DSF/Cu treatment leads to cell senescence and apoptosis in meningioma cells. Moreover, our Ingenuity Pathway Analysis revealed that the transcription factors ATF3, ATF4, ATF6, and XBP1 were activated in meningioma cells exposed to DSF/Cu. These transcription factors are related to the unfolding protein response (UPR), which serves as a buffer to maintain the ER homeostasis in the face of stresses such as increased levels of oxidizing agents within the ER. We found that DSF/Cu, especially in MgACs, activated the PERK pathway. Rozpędek et al., had discussed that PERK-dependent UPR resulted in adaptive or apoptotic response via eIF2α [54], which is consistent with our results. The Ingenuity Pathway Analysis predicted eIF2 singling pathway would be related to DSF/Cu treatment. Moreover, we found that with a short DSF/Cu treatment period, MgACs revealed G1 arrest, which is the adaptive response. However, when the DSF/Cu treatment was prolonged, the ATF4 and CHOP were increased, which ultimately led to apoptosis. However, our data revealed that the Bax, cleaved caspase 3, and procaspase/cleaved caspase 9 revealed no expression change before and after DSF/Cu treatment. We thought this apoptotic effect in meningioma induced by DSF/Cu might be related to the extrinsic pathway.

ER stress is associated with autophagy, and our data showed that DSF/Cu induced autophagy in meningioma. However, the mechanism of ER stress and the interactions between ER stress, apoptosis, and autophagy are complicated [55,56,57]. Further experiments will explore the crosstalk between apoptosis and autophagy induced by DSF/Cu in meningioma. Moreover, under ER stress, inter-organellar crosstalk occurs between the ER and mitochondria through direct contact via the mitochondria-associated membrane (MAM) [58], which is linked to the outflow of Ca^2+^ from the ER to the mitochondria [59,60]. More experiments will be necessary to determine the impact of DSF/Cu on MAM function and Ca^2+^ mobilization in meningioma cells.

Acetaldehyde dehydrogenase (ALDH) is a family of enzymes that catalyzes the oxidation of acetaldehyde during ethanol metabolism [61]. Nineteen genes encode different ALDH isotypes in the human genome. Many studies have noted that cancer cells exhibiting high ALDH expression (e.g., glioblastoma) also show enhanced tumor formation and resistance to chemotherapy [62,63,64]. ALDH was regarded as a functional marker in cancer stem cells because it serves as an essential metabolic enzyme involved in gene expression, protein translation, and signal transduction [65]. In gliomas, the elevation of ALDH 1A1 and 1A3 is related to high cell invasiveness and poor survival outcome [66,67]. We found that DSF/Cu suppressed different ALDH isoforms in MgACs and in MgSCs, and that effect was more pronounced in MgSCs than MgACs. In addition, DSF/Cu inhibits ALDH activity in glioblastoma stem-like cells, which has a cytotoxic effect [68]. The effect of DSF/Cu on ALDH activity in meningioma needs more experiments to explore. On the other hand, Paranjpe et al., reported that DSF increases the sensitivity of glioblastoma to chemotherapy through inhibition of human O6-methylguanine-DNA methyltransferase [15]. Whether DSF/Cu can enhance the effects of chemotherapy and radiation therapy on meningioma remains to be determined.

## 5. Conclusions

This is the first study to explore the effect of DSF/Cu on meningioma. Our findings demonstrate the cytotoxicity of DSF/Cu against meningioma cells and stem-like cells. We believe DSF/Cu is a potentially practical approach in treating meningioma.

## Figures and Tables

**Figure 1 biomedicines-10-00887-f001:**
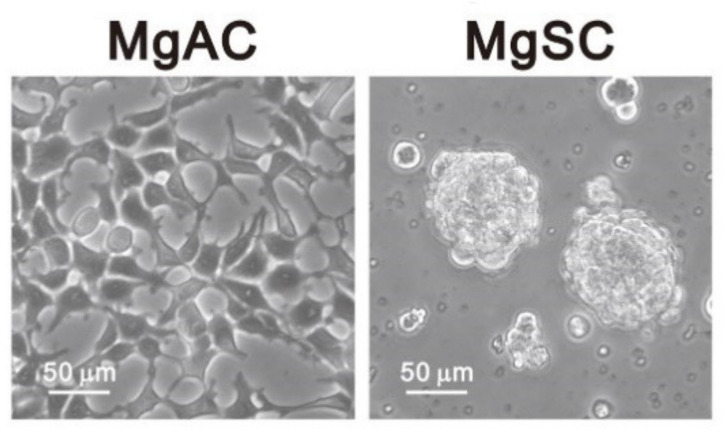
Morphology of adherent meningioma cells (MgACs) and meningioma sphere cells (MgSCs). For MgSCs, we collected MgACs and cultured them for 2 days in DMEM/F12 (Corning, Glendale, AZ, USA) supplemented with N2 (Thermo Fisher Scientific, Waltham, MA, USA), epidermal growth factors (EGF, PeproTech Asia, Rehovot, Israel), and fibroblast growth factors (FGF, PeproTech, Asia, Rehovot, Israel) to form the sphere cells. Scale bar: 50 μm.

**Figure 2 biomedicines-10-00887-f002:**
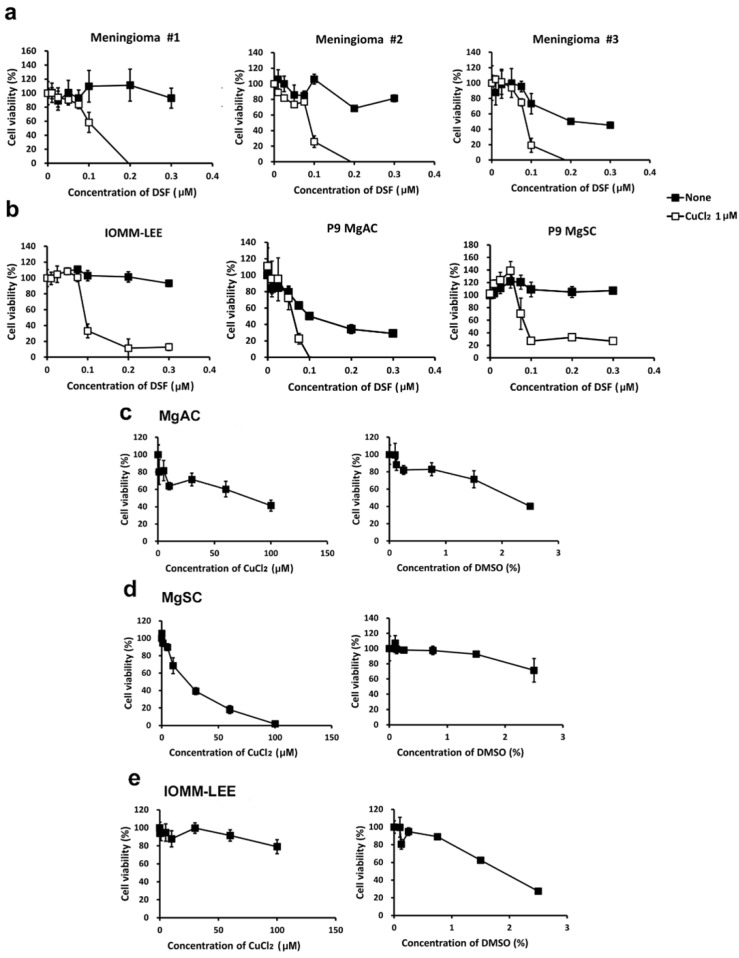
DSF/Cu reduced the viability of meningioma cells. Meningioma #1, #2, and #3 (**a**); IOMM-Lee; MgACs; and MgSCs (**b**) were incubated for 48 h with selected concentrations of disulfiram (DSF, 0 μM, 0.01 μM, 0.025 μM, 0.05 μM, 0.075 μM, 0.1 μM, 0.2 μM, and 0.3 μM), with or without 1 μM CuCl_2_. (**c**–**e**) MgACs, MgSCs, and IOMM-Lee were incubated for 48 h with selected concentrations of Cu^2+^ (CuCl_2_) (0 μM, 0.1 μM, 1 μM, 5 μM, 10 μM, 30 μM, 60 μM, and 100 μM) or DMSO (control) (0%, 0.1%, 0.125%, 0.25%, 0.75%, 1.5%, and 2.5%). Cell viability was assessed with MTS assays. Results are presented as the mean ± SD of triplicate samples from representative data from three independent experiments.

**Figure 3 biomedicines-10-00887-f003:**
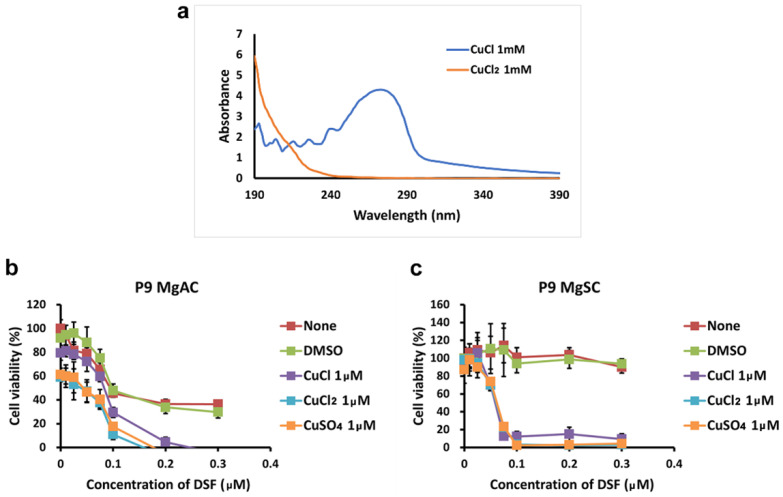
Effect of copper ion valence on the cytotoxicity of the DSF/Cu complex (**a**) UV-visible absorption spectra of CuCl and CuCl_2_ in aqueous solution with a concentration of 1 mM. MgACs (**b**) and MgSCs (**c**) were incubated for 48 h with different concentrations of DMSO (0 μM, 0.01 μM, 0.025 μM, 0.05 μM, 0.075 μM, 0.1 μM, 0.2 μM, and 0.3 μM) or DSF (0 μM, 0.01 μM, 0.025 μM, 0.05 μM, 0.075 μM, 0.1 μM, 0.2 μM, and 0.3 μM) with or without CuCl, CuCl_2_, or CuSO_4_. Cell viability was assessed with MTS assays. Results are presented as the mean ± SD of triplicate samples from representative data from three independent experiments.

**Figure 4 biomedicines-10-00887-f004:**
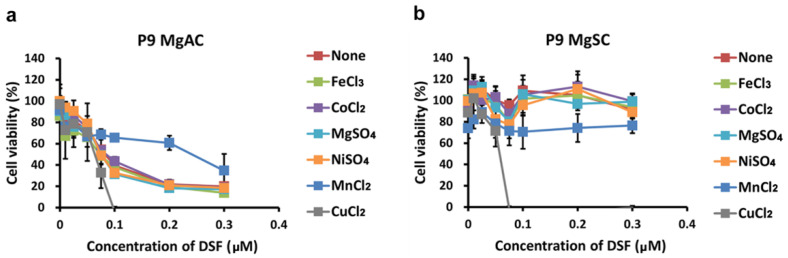
Effect of various metal ions on the cytotoxicity of the DSF/metal ion complex. MgACs (**a**) and MgSCs (**b**) were incubated for 48 h with different concentrations of DSF (0 μM, 0.01 μM, 0.025 μM, 0.05 μM, 0.075 μM, 0.1 μM, 0.2 μM, and 0.3 μM) and 1 μM FeCl_3_, CoCl_2_, MgSO_4_, NiSO_4_, or MnCl_2_. Cell viability was assessed with MTS assays. Results are presented as the mean ± SD of triplicate samples from representative data from three independent experiments.

**Figure 5 biomedicines-10-00887-f005:**
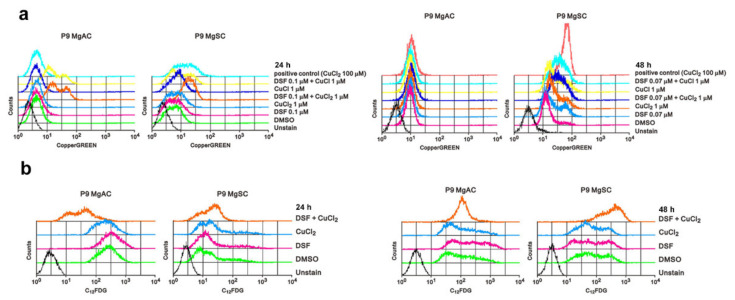
DSF/Cu induces intracellular copper accumulation and cell senescence in meningioma cells. (**a**) Cells were incubated for 24 h and 48 h with the indicated concentrations of DSF and Cu alone and in combination. Intracellular Cu^+^ concentrations were measured using CopperGREEN dye and analyzed with flow cytometry. Cells without CopperGREEN served as blanks. (**b**) Cells were incubated for 24 h or 48 h with 0.1 μM DSF with or without 1 μM CuCl_2_, after which they were stained with C_12_FDG and assayed using a flow cytometer. Cells without C_12_FDG served as blanks.

**Figure 6 biomedicines-10-00887-f006:**
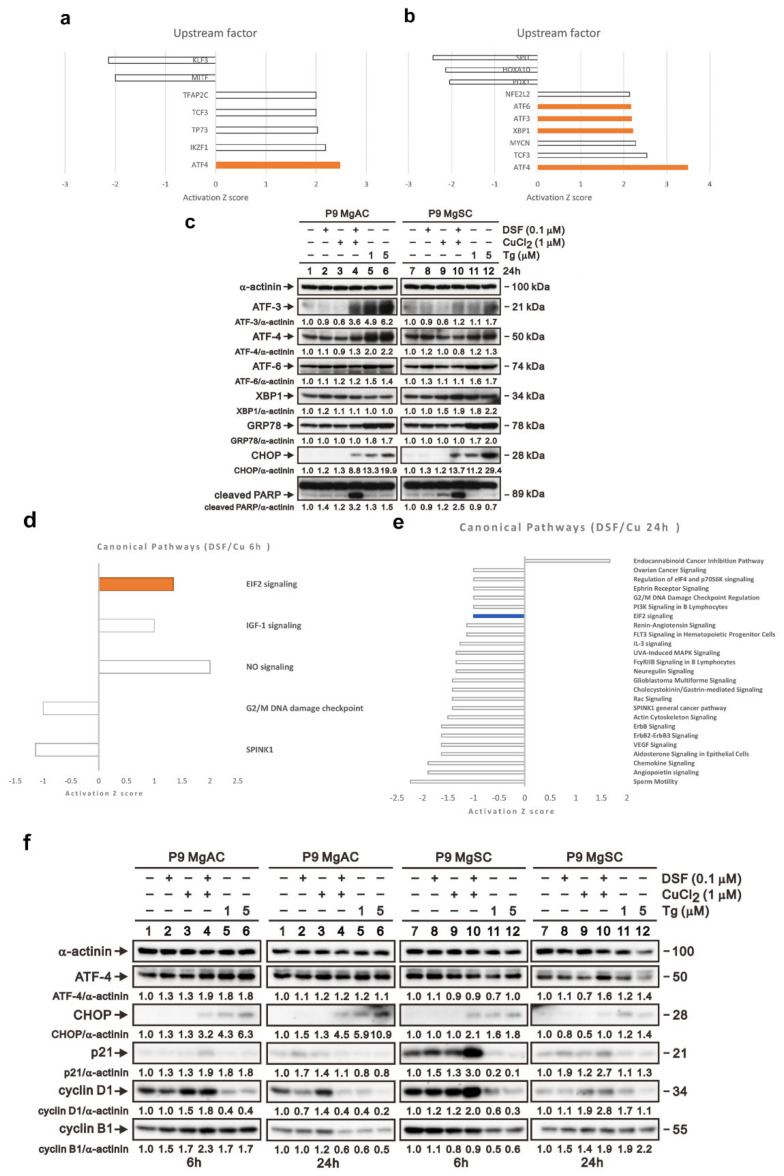
DSF/Cu induced ER stress in meningioma cells. The bar chart shows potential upstream regulators predicted with Ingenuity Pathway Analysis based on microarray data from MgACs treated for 6 h (**a**) or 24 h (**b**) with DSF and 1 μM CuCl_2_. Shown are upstream regulators for which there was at least a 1.3-fold change in expression compared to vector control cells. (**c**) Cells were incubated for 24 h with 0.1 μM DSF and 1 μM CuCl_2_ alone or in combination, after which expression of proteins related to the unfolded protein response was confirmed by Western blotting. Thapsigargin (Tg) is an inducer of the ER stress and served as a positive control; α-actinin served as a loading control. The quantitative data normalized with internal control are labeled below each protein band. The bar chart shows potential canonical pathways predicted with Ingenuity Pathway Analysis based on microarray data from MgACs treated for 6 h (**d**) or 24 h (**e**) with DSF and 1 μM CuCl_2_. Shown are upstream regulators for which there was at least a 1.0-fold change in expression compared to vector control cells. (**f**) Cells were incubated for 24 h with 0.1 μM DSF and 1 μM CuCl_2_ alone or in combination, after which expression of proteins related to the unfolded protein response was confirmed by Western blotting. Thapsigargin (Tg) is an inducer of the ER stress and served as a positive control; α-actinin served as a loading control. The quantitative data normalized with internal control are labeled below each protein band.

**Figure 7 biomedicines-10-00887-f007:**
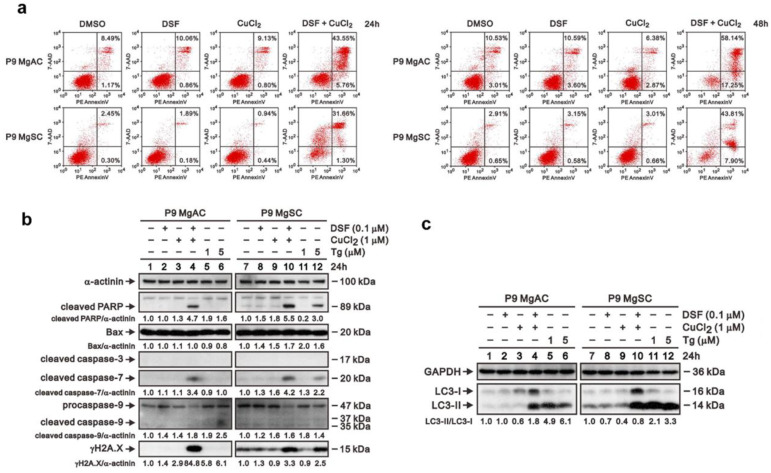
DSF/Cu induced cell apoptosis and autophagy in meningioma cells. (**a**) Cells were incubated for 24 h or 48 h with 0.1 μM DSF with or without 1 μM CuCl_2_. Cell apoptosis was analyzed based on flow cytometric analysis of Annexin V staining. Cells were incubated for 24 h with 0.1 μM DSF and 1 μM CuCl_2_ alone or in combination, after which expressions of proteins related to apoptosis (**b**) and autophagy (**c**) were evaluated by Western blotting. Thapsigargin (Tg) served as a positive control; α-actinin served as a loading control. The quantitative data normalized with internal control are labeled below each protein band.

**Figure 8 biomedicines-10-00887-f008:**
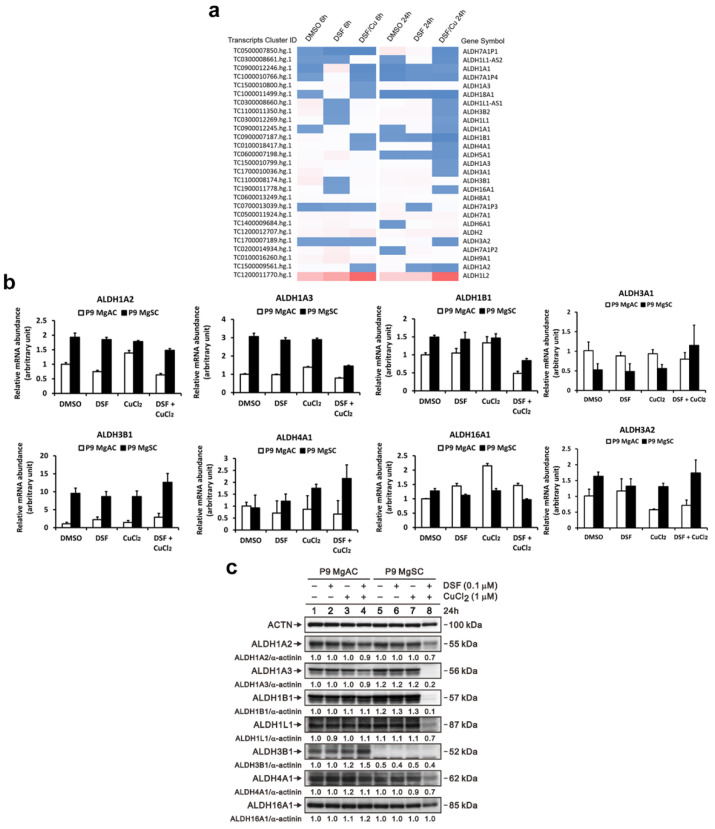
DSF/Cu inhibits different ALDH isoforms’ expressions in MgACs and MgSCs. (**a**) Heatmap showing normalized gene expression from a microarray database analysis and highlighting significantly up- and downregulated ALDH isoforms in MgACs treated for 6 h or 24 h with DMSO, 0.1 μM DSF, or 0.1 μM DSF/1 μM CuCl_2_. Expression of ALDH isoforms was verified using qRT-PCR (**b**), and Western blot (**c**) with MgACs and MgSCs incubated for 24 h with 0.1 μM DSF and 1 μM CuCl_2_; α-actinin served as a loading control. The quantitative data with normalized internal control are labeled below each protein band.

**Table 1 biomedicines-10-00887-t001:** Sequences of primers used in the polymerase chain reaction.

Gene Name	Primer Sequence (5′->3′)
ALDH1A2	Forward: 5′-TTGCAGGGCGTCATCAAAAC-3′
	Reverse: 5′-ACACTCCAATGGGTTCATGTC-3′
ALDH1A3	Forward: 5′-TGAATGGCACGAATCCAAGAG-3′′
	Reverse: 5′-CACGTCGGGCTTATCTCCT-3
ALDH1B1	Forward: 5′-AGAGTCTTACGCCTTGGACTT-3′
	Reverse: 5′-GTCTTGCCATGCCACTTGTC-3′
ALDH3A1	Forward: 5′-CTCGTCATTGGCACCTGGAACT-3′
	Reverse: 5′-CTCGCCATGTTCTCACTCAGCT-3′
ALDH3A2	Forward: 5′-ACTGATAGGAGCCATCGCTGCA-3′
	Reverse: 5′-GCTCCGTGGTTTCCTCAACACC-3′
ALDH3B1	Forward: 5′-GCCCTGGAACTATCCGCTG-3′
	Reverse: 5′-CGTTCTTGCTAATCTCCGATGG-3′
ALDH4A1	Forward: 5′-GACGTGCAGTACCAAGTGTC-3′
	Reverse: 5′-GATCACGGTCTTACCCTGTCC-3′
ALDH16A1	Forward: 5′-CTGCTCCACTACCATGCAATCC-3′
	Reverse: 5′-GCAGGGCAAATCCTCCACATCA-3′

## Data Availability

Not applicable.

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
