# Peer review of "The Effect of Disulfiram and Copper on Cellular Viability, ER Stress and ALDH Expression of Human Meningioma Cells"

_biomedicines, 2022, doi:10.3390/biomedicines10040887_

Round 1

Reviewer 1 Report

Well written introduction, clear purpose of the study, wisely chosen material and methods to pursue results. Adequate statistical analysis. Very interesting results. Innovative conclusions because they open new possible therapeutic avenues in anaplastic meningiomas to improve the quality of life of patients, to reduce relapses and to increase disease-free survival when they are developed. Your study represents an invitation to continue on this path of research with the hope of moving from a pre-clinical phase to a clinical one as soon as possible. 

Author Response

Reviewers’ comments:

Reviewer 1

Well written introduction, clear purpose of the study, wisely chosen material and methods to pursue results. Adequate statistical analysis. Very interesting results. Innovative conclusions because they open new possible therapeutic avenues in anaplastic meningiomas to improve the quality of life of patients, to reduce relapses and to increase disease-free survival when they are developed. Your study represents an invitation to continue on this path of research with the hope of moving from a pre-clinical phase to a clinical one as soon as possible.

Response: We sincerely appreciate the reviewer’s comment. It is hoped that the results of our research can be applied to clinical treatment in the future to improve the prognosis and the quality of life of patients with malignant meningioma. We thank the reviewer’s comment.

Reviewer 2 Report

In the present manuscript, the authors demonstrated the effects of disulfiram  + copper on meningioma cells.

The title should be improved as that showing the results of the present study more clearly, not "repurposing".

There were some typing errors in the present manuscript.

Line 69, "by Stupp et al., with RTOG 0525 and RTOG 0825 [18-25]", the reference [18] was by Stupp et al., but the references [19] and [20] were not by Stupp et al. The text should be revised more accurately.

In Materials and Methods, 2.1., the explanation of cell culture was not easy to understand. Primary culture of meningioma cells from patients (#1, #2, and #3), P9 MgACs and P9 MgSCs, MgACs, MgSCs, and IOMM-Lee cells were used in the present study. The cells other than IOMM-Lee cells were not clearly distinguished. Moreover, in line 100 "To obtain MgSCs sphere cultures, MgACs were maintained in...."; however, MgACs were daughter cells of MgSCs in the reference [21]. The explanation of 2.1. was confused.

In Results, line 205-208, if the authors showed "Cu+ and Cu2+ were not as cytotoxic alone as in combination with DSF", the experiments comparing the cytotoxicity with or without DSF were needed in Figure 2c, 2d, and 2e.

In Results 3.4. and Figure 5, "the cytotoxic activity of DSF was Cu-dependent", the authors described DSF/Cu increased intracellular Cu+ levels and the cellular senescence. However, it was not clear in figure 5 whether DSF/Cu group only increased. The chart looked similar in DMSO, DSF, Cu, DSF+Cu groups. In addition, why was "none" group as negative control absent?

In the results of western blotting, Figure 6c, 6f, 7b, 7c, and 8c; the quantitative data should be added if the authors expressed 'increased' or 'decreased'.

In Figure 8, and the text line 326-333; why did the authors write the changes in expression of ALDH 1A2, 1A3, 1B1 only? The expressions of ALDH 3A1, 3B1, 4A1 might be increased with DSF/Cu in MgSCs as shown in the graphs. Moreover, ALDH 1A2 and 1A3 were not clearly decreased in MgACs. Therefore, the authors should present the data more precisely. 

Reviewer 3 Report

Finding a good drug to treat malignant brain tumors would be excellent, and repurposing disulfiram has attracted a lot of interest. However, most papers and all manuscript I have reviewed lacked a clear hypothesis and the research sometimes appears as being conducted more by the available methods than a good hypothesis for its mode of action.

This basic critique appears to be relevant for this manuscript, too. Thapsigargin is used as a positive control, its main mode of action has been described as inhibiting Ca outflow of the ER, and other effects follow this primary effect. If this mechanism is seen as crucia than the impact of DSF/Cu on MAM integrity should be clarified first before drawing conclusions based on this assumption as a fact in the discussion.

Similarly; high expression of ALDH isoenzymes are associated with chemotherapy resistance. However, I do see an increase in most ALDH mRNA tests which should result in an increased cdrug resistance, since high ALDH levels are associated with malignancy. Only some isoenzymes are expressed with less protein in SC but not in AC cells, and I do not see a coherent interpretation for these discrepancies, both high mRNA and no changes or less protein within their experiments, as well as high expression and increased resistance versus DSF sensitizing effects.

A final point is worth mentioning. The experiments are performed with DSF in cell cultures; in vivo DSF is quickly metabolized to dithiocarb. Since this is the main metabolite why didn't the authors use this compound for their experiments?

Some specific comments:

p. 7: I don‘t understand why UV spectra are included since they do not contribute to the effects and interpretation.

p.7: Figure 3b indicates cell viability by DMSO at 0,3 µM to be roughly 35%; from Figure 2b I read a viability of ~50%. Please explain.

p. 7, Figure 4: For P9 MgAC cells 0,3 µM DSF ("none") has still another value for cell viability, namely 20%.

p. 8, Figure 5: Flow cytometry usually creates scatter plots; I do not see what the authors have measured to obtain these spectrum-like plots, especially not with an abscissa label like „CopperGREEN“. No ordinate label is given so I cannot interpret this figure.

Figure 6: I do not share the interpretation of the authors. Tg is supposed to be a positive control for unfolding protein response; this line therefore serves as an indicator of cell damage due to high cytosolic Ca. In Fig. 6F, its response only coincides with CHOP, not even with ATF-4 (with a larger decrease (in my eyes) at 1 µM compared to 5 µM). Thus, a decrease in cyclin proteins I can rationalize but not protein unfolding stress by DSF/Cu. For clarity, the authors should also provide lanes for ATF3 and 4 in Fig. 6F to support their claim.

p. 10, Figure 7: I know 7-AAG as 7-Amino-actinomycin-D, an intercalator for GC-rich DNA. Is this also true in the presented experiments? The compound is supposed to separate between cells in different stages of its cycle, or going into apoptosis. A reference for the fact that the observed changes in FACS distribution indeed are due to apoptosis but not other cell cycle steps is necessary (the only reference in the methods section is a self citation).

p. 11: The authors introducec ALD isoenzymes without a rationale for their determination. This isoenzyme is not mentioned in the Introduction and thus appears disconnected.

Round 2

Reviewer 2 Report

Line 16 in Abstract, "comment" might be error of "common".

Figure 2b (P9 MgSC) in the revised manuscript was different from that in the original. Which was correct data?